# Numerical investigation of seismic amplification characteristics in loess ridge region of Xiji, northwest China

Da Peng[1]*, Jingshan Bo[1,2,3], Chaoyu Chang[2,3], Wenhao Qi[1], Xiaobo Li[2,3]

1 Key Laboratory of Earthquake Engineering and Engineering Vibration, Institute of Engineering Mechanics, China Earthquake Administration, Harbin, China, 2 Hebei Key Laboratory of Earthquake Disaster Prevention and Risk Assessment, Sanhe, China, 3 Institute of Disaster Prevention, Institute of Geological Engineering, Sanhe, China

* pengda1992@gmail.com

**Data Availability Statement:** All relevant data are within the paper and its Supporting Information files.

**Funding:** This work is financially supported by the National Natural Science Foundation of China (No. U1939209),Scientific Research Fund of Institute of

## Abstract

The seismic effects on sloped terrain, which are of paramount importance for engineering design and earthquake risk mitigation, have always been a central focus of earthquake engineering research. In this study, generalized geometric models of loess ridges at varying heights were created, and a three-dimensional nonlinear numerical model was established using FLAC3D. Seismic ground motion time histories at different frequencies and actual earthquake ground motion records were input into the model to analyze the peak acceleration amplification effects experienced by the surface of loess ridges when subjected to SV waves. The study's outcomes reveal that seismic amplification on the slopes of loess ridges is characterized by non-linearity with respect to slope height. Instead, it exhibits rhythmic variations, with the rate of change in these rhythms increasing in correspondence with the frequency of seismic motion and the height of the slope. Under low-intensity seismic motion, a linear increase in acceleration amplification is observed at the ridge's crest concerning the height of the loess ridge. However, under high-intensity seismic motion, the relationship between amplification and slope height becomes less significant. Typically, the peak acceleration at the ridge's crest is reported to be 1.5 to 2.5 times that observed at the slope's base. The amplification effect at the ridge's crest is more pronounced in the low-frequency and high-frequency segments when compared to the mid-frequency range. Conversely, significant amplification is observed in the high-frequency range in the lower sections of the slope near the base. It is further noted that the amplification effect at the ridge's crest displays distinct behavior at different frequencies, characterized by narrow frequency bands of maximum amplification, with peak amplification factors exceeding 10 in some cases. These research findings have practical significance and provide valuable references for engineering construction and seismic risk mitigation planning in loess regions.

Engineering Mechanics, China Earthquake
Administration (Grant NO.2020EEEVL0201).

**Competing interests:** The authors have declared
that no competing interests exist.

## 1. Introduction

The terrain of slopes has been widely recognized as having a significant influence on seismic
characteristics. Differences in seismic damage at various locations on slopes have been
observed in several earthquakes, such as the 1994 Northridge Earthquake [1], the 2005 Kash-
mir Earthquake [2], the 2008 Wenchuan Earthquake [3], and the 2010 Haiti Earthquake [4].
Notably, the damage to buildings at the crest during these earthquakes was significantly greater
than the damage observed at lower elevations. Understanding the seismic effects of sloped ter-
rain is of paramount importance for earthquake risk reduction.

The seismic amplification of sloped terrain is a central concern in the study of the seismic
effects of such terrain. Many researchers have investigated and summarized the amplification
of seismic waves in sloped terrain through field observations and measurements [5–9], analyti-
cal solutions [10–13], physical model experiments [14–16], and numerical simulations [17–
21]. Sloped terrain can be classified into single-sided and double-sided slopes based on their
shape. A single-sided slope has one side with sloping terrain, while the other side may be flat
or lack steep slopes. The non-symmetric geometry of single-sided slopes complicates analytical
solutions and site-specific numerical simulations, making their conclusions difficult to gener-
alize [22, 23]. Double-sided slopes, on the other hand, feature sloping terrain on both sides
and are often studied in the context of ideal rock slopes. The propagation paths, scattering,
and interference of seismic waves on double-sided slopes are notably different from those on
single-sided slopes, leading to rhythmic variations in seismic amplification on the slope surface
[19, 23].

The Loess Plateau is located in the mid to upper reaches of the Yellow River in northwest
China, primarily encompassing provinces such as Gansu, Shaanxi, Shanxi, and the Ningxia
Hui Autonomous Region. It spans an area of approximately $6.3×10^5$ km$^2$, characterized by an
overall topography of higher elevations in the west gradually descending to lower elevations in
the southeast. The average elevation decreases from around 2000 m in the west to 1000 m in
the southeast. In this region, a unique climate has led to the widespread distribution of exten-
sive, continuous, and thick loess deposits. Over the past two million years, the interplay of
loess accumulation, erosion, and transportation, influenced by both internal and external geo-
logical forces, has given rise to the distinctive modern landforms of the Loess Plateau, includ-
ing loess tableland, loess ridges, and loess hills. The term " Loess Ridge" or " Loess Double-
Sided Slope" refers to a characteristic landform found in the Loess Plateau region. It is a long
and narrow landform primarily composed of loess soil. Loess ridges exhibit a relative height
difference ranging from 50 m to 200 m, with a relatively flat, arched top, as shown in Fig 1.
Due to the unique origin and structural characteristics of loess, the dynamic properties of loess
are complex, resulting in seismic amplification effects distinct from those in rocky or other soil
slopes [24, 25]. In several seismic events [26, 27], significant seismic damage has been observed
on loess slopes, particularly in regions distant from the earthquake epicenters. In the 2008
Wenchuan Ms8.0 Earthquake, loess areas experienced considerable damage, with clear indica-
tions of seismic amplification and localized ground motion amplification [9]. Loess ridges
serve as important sites for village communities and engineering projects in the Loess region,
making it essential to understand their seismic amplification characteristics to guide engineer-
ing design effectively.

In this study, the model focused on the region of Xiji (Fig 1A), located in south of Ningxia
Autonomous Region and east of Gansu Province, a seismic-prone area in the western Loess
Plateau. The loess in this region demonstrates uniform thickness, with relatively minor varia-
tions in physical and mechanical properties. This work involves the creation of geometric
models for loess ridges with varying heights. It utilizes in-situ testing and laboratory

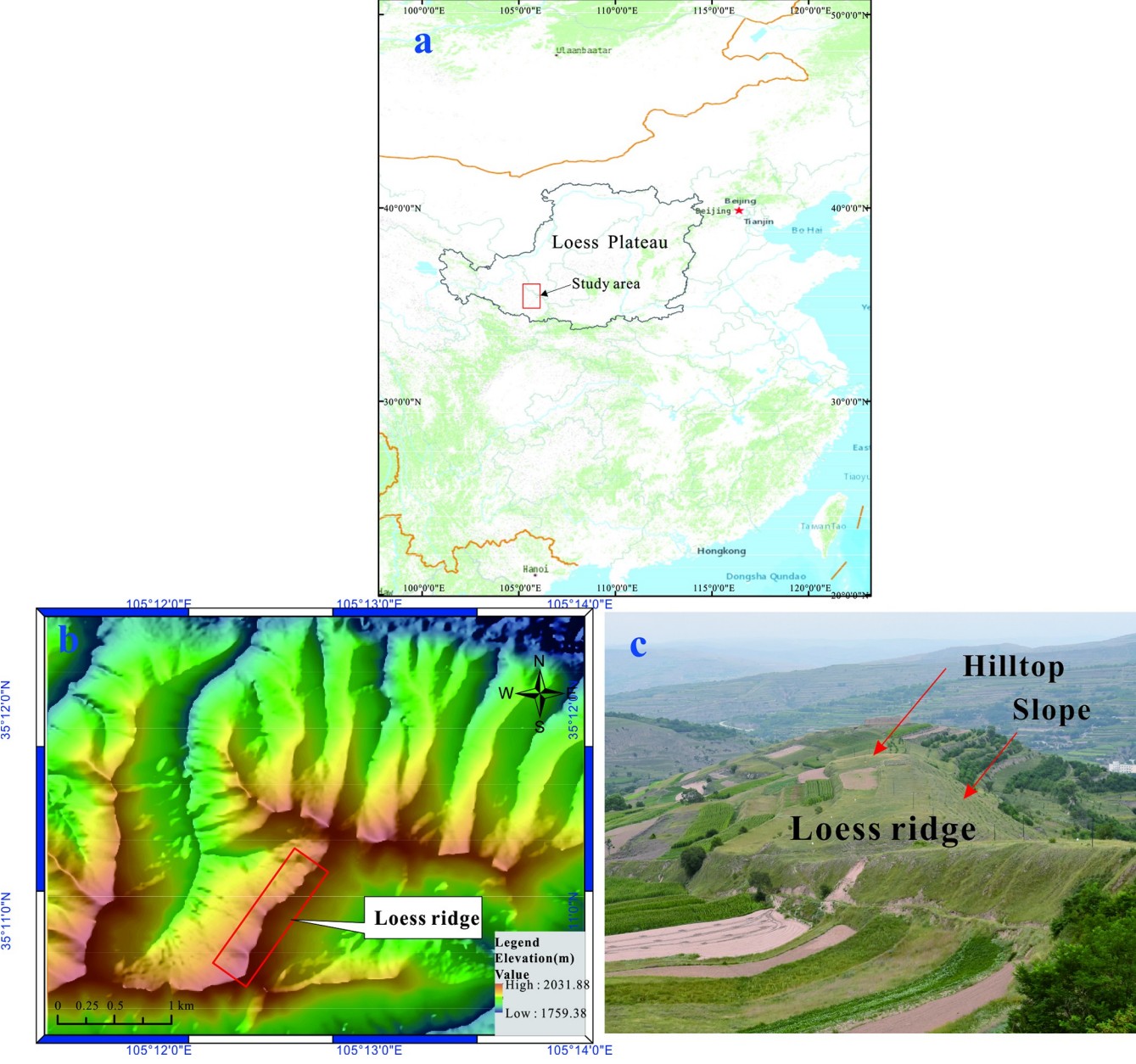

**Fig 1.** a. Location of Loess Plateau b. Topographic map of loess ridges Area, c Photo of loess ridge.

experiments to obtain dynamic parameters for typical loess soil. Leveraging finite difference numerical methods [28], a three-dimensional nonlinear numerical model of loess ridges is established. Various frequency dynamic time histories and actual earthquake ground motion records are input into the model to analyze the peak acceleration amplification effects on the surface of loess ridges when subjected to SV waves. This research explores the seismic acceleration amplification characteristics of the slope surfaces of loess ridges, considering different slope heights. The impact of slope height and seismic motion frequency on acceleration amplification was analyzed, providing insights into the amplification effects on the crest of loess ridges under different seismic conditions.

## 2. Materials and methods

### 2.1 loess ridge model

Loess ridges can be classified into three types based on their origin: inherited, erosional, and mixed. Among these, the predominant type in the earthquake-prone Longxi region of China is inherited loess ridges [29, 30]. Inherited loess ridges have an initial bedrock terrain with a double-sided slope, and windblown loess layers are subsequently deposited on the initial slope, resulting in a loess double-sided slope with a shape and slope similar to the original bedrock terrain. This layering structure is also referred to as the loess-mudstone binary slope structure.

In this study, 3D models of loess ridges were created, and seismic response analysis was conducted using the finite difference software FLAC3D 7.0. Fig 2 illustrates the analyzed slope geometry. The relative height difference (H) between the slope shoulders and the toe of the slope was considered, with H values ranging from 50m to 150m, based on statistical data for loess ridge heights. The slope shoulders extend smoothly as ellipses from the crest, and the crest itself is arc-shaped, maintaining a fixed height difference of 10m from the slope and arc transition endpoints. The height difference from the slope toe to the crest is denoted as $H_{max}$. The distance between the toes of the double-sided slope is 700m.

To minimize artificial wave reflections at the model boundaries, a horizontal extension of 600m was added along the slope toe location of the model. The bottom of the model had static boundaries, while free-field boundaries were applied laterally. These boundary conditions effectively absorb outward-propagating energy and reduce wave reflections. Furthermore, the free-field boundary conditions allow for the simulation of 2D and 1D wave propagation [31, 32].

The structural parameters of the soil layers are based on drilling and shear-wave velocity test results conducted on the slopes in loess regions, as shown in Fig 3. The loess double-slope model is treated as a binary structure consisting of loess and mudstone. The overlying layer consists of 40m-thick loess, which has been divided into four layers based on the shear-wave velocity characteristics and laboratory test results. These layers are defined as follows: 0-5m (layer 1), 5-15m (layer 2), 15-25m (layer 3), and 25-40m (layer 4). The lower part of the model consists of firm mudstone.

To account for the non-linear behavior of loess, an elastoplastic constitutive model with a Mohr-Coulomb yield criterion is employed for modeling the soil layers. The soil parameters are derived from laboratory tests and are presented in Table 1.

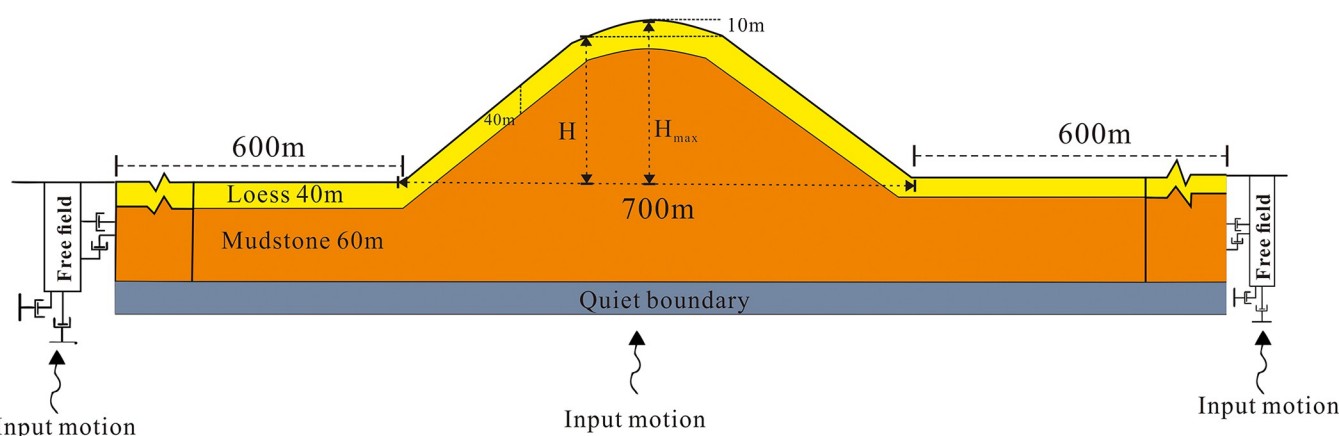

**Fig 2. Schematic illustration of model for the numerical analyses of loess ridges.**

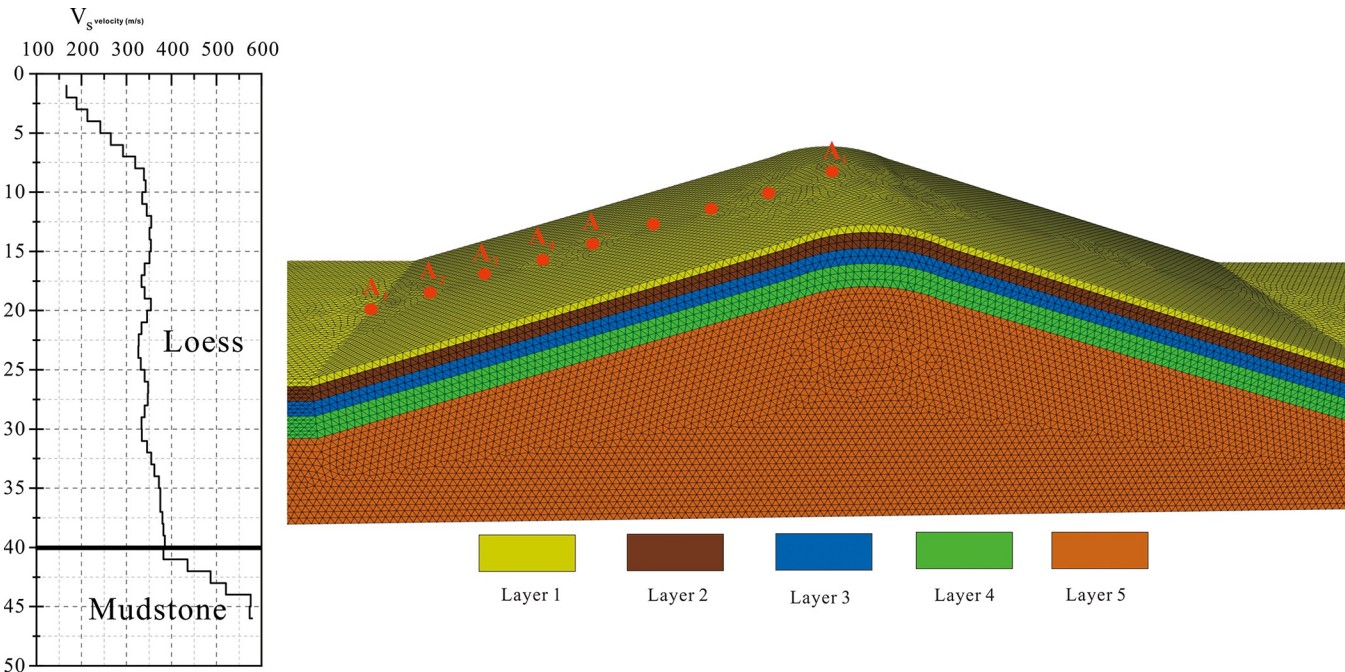

**Fig 3. Shear wave velocity structure chart and FLAC3D model of loess ridge.**

The grid size of the 3D loess ridge model ranges from 2 to 5 m. To ensure that waves with frequencies up to 10 Hz propagate through the grid without numerical distortion, the height of each zone is set to be smaller than 1/10 of the minimum wavelength in the corresponding medium [33]. In total, the model is divided into 507,744 (for H = 50m) to 4,489,025 (for H = 150m) hexahedral zones.

For monitoring the seismic acceleration response on the slope's surface, as shown in Fig 3, adjacent acceleration monitoring points ($A_1$, $A_2$, $A_3$,. . ., $A_{i-1}$, $A_i$) are set along the slope, with an elevation difference of 10 m. $A_1$ is located at the slope toe, $A_{i-1}$ represents the slope shoulder, and $A_i$ corresponds to the crest.

## 2.2 Input motion

In this study, two types of acceleration time history records were employed as input. One is a single-frequency input seismic motion time history known as Chang's signal [22, 34], which allows for the adjustment of frequency and amplitude. It is represented by Eq (1):

$$a(t) = \sqrt{\beta e^{-\alpha t} t^{\gamma} \sin (2\pi f t)} \tag{1}$$

**Table 1. Soil properties of each layer in the FLAC3D model.**

| No. | Materials | Thickness(m) | $V_S$ (m/s) | density (kg/m³) | K (MPa) | E (MPa) | φ (˚) | c(kPa) |
|---|---|---|---|---|---|---|---|---|
| Layer1 | Loess | 5 | 200 | 1210 | 105 | 48.4 | 25 | 25 |
| Layer2 | Loess | 10 | 300 | 1210 | 240 | 109 | 25 | 45 |
| Layer3 | Loess | 10 | 360 | 1349 | 390 | 175 | 28 | 60 |
| Layer4 | Loess | 15 | 350 | 1525 | 410 | 187 | 28 | 35 |
| Layer5 | Mudstone | | 600 | 2200 | 1900 | 792 | 160 | 35 |

K: Bulk Modulus, E: Shear Modulus, φ: Angle of internal friction, c: Cohesion

where $\alpha$, $\beta$, and $\gamma$ are constants controlling the shape and amplitude of the acceleration-time history, the seismic input comprises two types of ground motions with peak accelerations of 0.1 m/s$^2$ for low-intensity seismic motion and 1 m/s$^2$ for high-intensity seismic motion. The frequency (f) of the seismic input time history is taken at 1 Hz, 2 Hz, 5 Hz, and 10 Hz, and the duration (t) of the acceleration time history is set at 4 seconds. The acceleration time history records and the corresponding Fourier spectrum are depicted in Fig 4A and 4B.

Additionally, another set of seismic input time history record was obtained from the mainshock's north-south acceleration time history recorded at the Minxian Station, located 17.44 km from the epicenter of the Ms6.6 earthquake that occurred in the loess region of Gansu Province, China, on July 22, 2013 [35]. The original acceleration time history had a peak acceleration of 1.722 m/s$^2$, which was proportionally scaled to have a peak acceleration of 1 m/s$^2$ and a duration of 30 seconds. A 4th-order band-pass Butterworth filter was then applied to this time history, allowing frequencies between 0.1 Hz and 10 Hz to pass through. The baseline-corrected acceleration time history record is shown in Fig 4C. Fig 4D illustrates the Fourier amplitude spectrum of this record, revealing a bimodal frequency response with peaks at 4.3 Hz and 7.2 Hz.

To input seismic motion at a quiet boundary, a stress boundary condition is used (i.e., a velocity record is transformed into a stress record and applied to a quiet boundary [28]. The

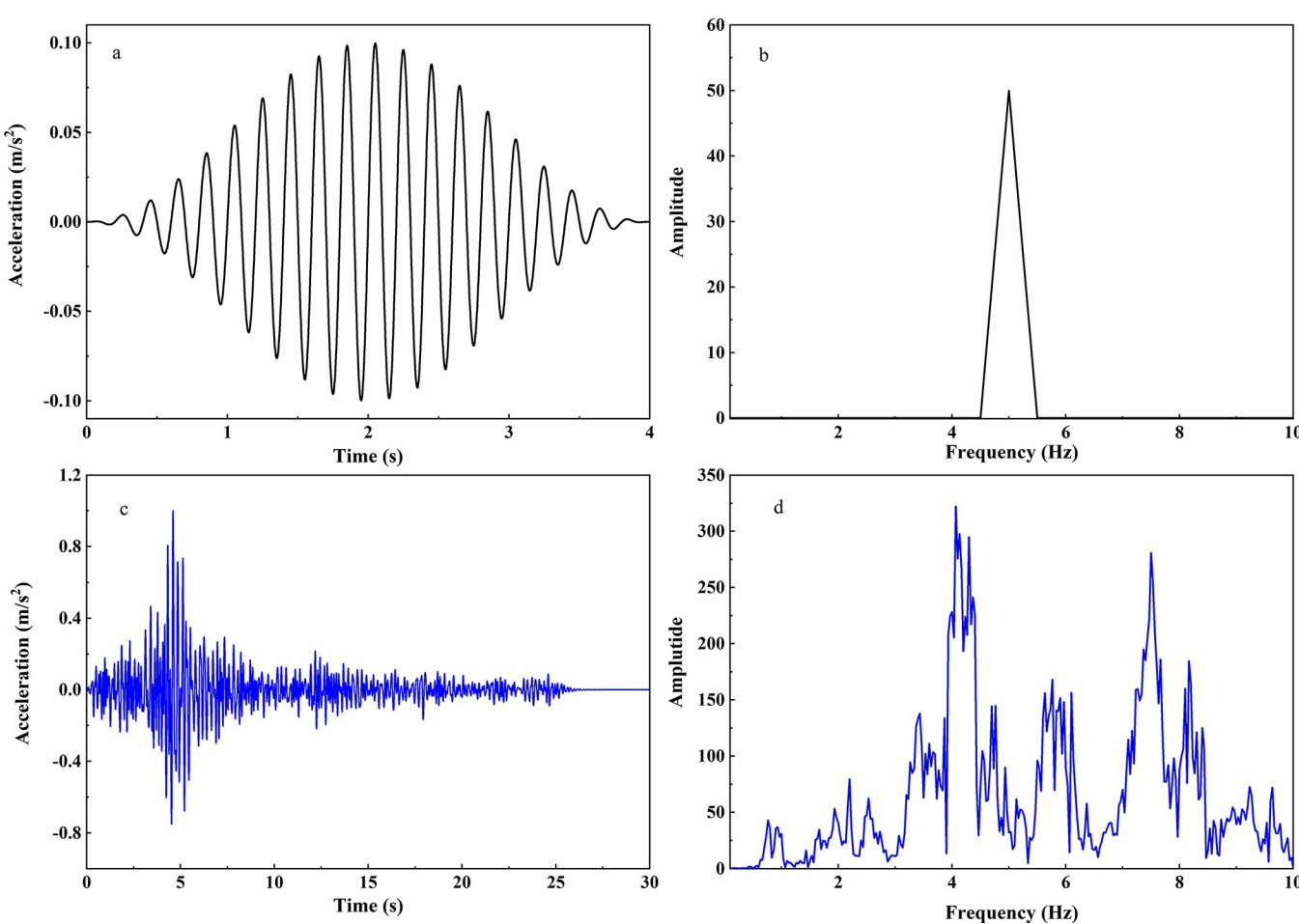

**Fig 4.** a. Schematic of Chang's signal, b. Fourier Amplitude Spectrum of Chang's signal c. Minxian seismic acceleration time History curve. d. Fourier Amplitude Spectrum of Minxian wave.

scaled acceleration is integrated to velocity, and the velocity time history v(t) is then converted to shear stress history $\sigma_s$(t) according to the following equation:

$$\sigma_s(t) = 2(\rho C_s)v(t) \tag{2}$$

Where, $\sigma_s(t)$ = applied shear stress history, $\rho$ = mass density, $C_s$ = speed of S-wave propagation through medium, $v(t)$ = input velocity history.

## 2.3 Nonlinear elastoplastic analysis

In FLAC3D, when using a simple elastoplastic constitutive model, the nonlinear dynamic characteristics of the soil can be incorporated by employing hysteresis damping. In this study, hysteresis damping is a key consideration for describing the nonlinear hysteresis behavior of the soil under seismic action. Hysteresis damping plays a role in characterizing the nonlinearity and damping properties of loess soils in seismic engineering, aiding in a more accurate simulation of the actual dynamic response of the soil. In our research, we conducted resonance column tests on loess samples at different depths (5m, 10m, 20m, 30m) to obtain the dynamic modulus and damping characteristics of the soil. The hysteresis damping was modeled using a three-parameter form, and through iterative adjustments, it was ensured to closely match the results of dynamic tests. This hysteresis damping model is material-independent and adopts the damping curve from dynamic tests, effectively capturing the dynamic nonlinearity of loess soils. Different layers of loess were characterized with respect to their dynamic shear modulus ratio and damping ratio through resonant column tests, as detailed in reference [36].

The shear modulus of the soil layers was determined through shear wave velocity tests. The dynamic shear modulus ratio was represented using the Sigmoidal model (sig3) within the hysteresis damping framework. For the different layers of loess, the parameters for the Sigmoidal model are as follows:

- For the layer1 of loess: a = 1.012, b = -0.465, x0 = -1.305.

- For the layer2 of loess: a = 1.012, b = -0.458, x0 = -1.25.

- For the layer3 of loess: a = 1.012, b = -0.4889, x0 = -1.05.

- For the layer4 of loess: a = 1.017, b = -0.567, x0 = -0.85.

These parameters help capture the nonlinear behavior of loess as the shear strain increases, allowing the dynamic shear modulus to decrease. The underlain mudstone layer in the loess is controlled by Rayleigh damping to account for the high-frequency seismic effects. The minimum damping ratio and central frequency for Rayleigh damping are set to 0.5% and 3 Hz, respectively.

Fig 5 demonstrates the calibration of numerical simulations with test results, showing how hysteresis damping effectively represents the attenuation of dynamic shear modulus in loess with increasing shear strain. Rayleigh damping is used to control the influence of high-frequency seismic motion in the underlying mudstone layer.

## 3. Results and discussion

The elevation difference between $A_{(2,3...i-1)}$ at different locations and $A_1$ at toe on the loess ridges, relative to the slope's height H, is represented by the elevation difference ratio η. Based on the results of a numerical model analysis, acceleration time histories and peak ground accelerations were obtained at $A_1$ and $A_{(2,3...i-1)}$ locations. An elevation amplification coefficient ε is introduced to account for variations in the slope surface elevation and is defined as

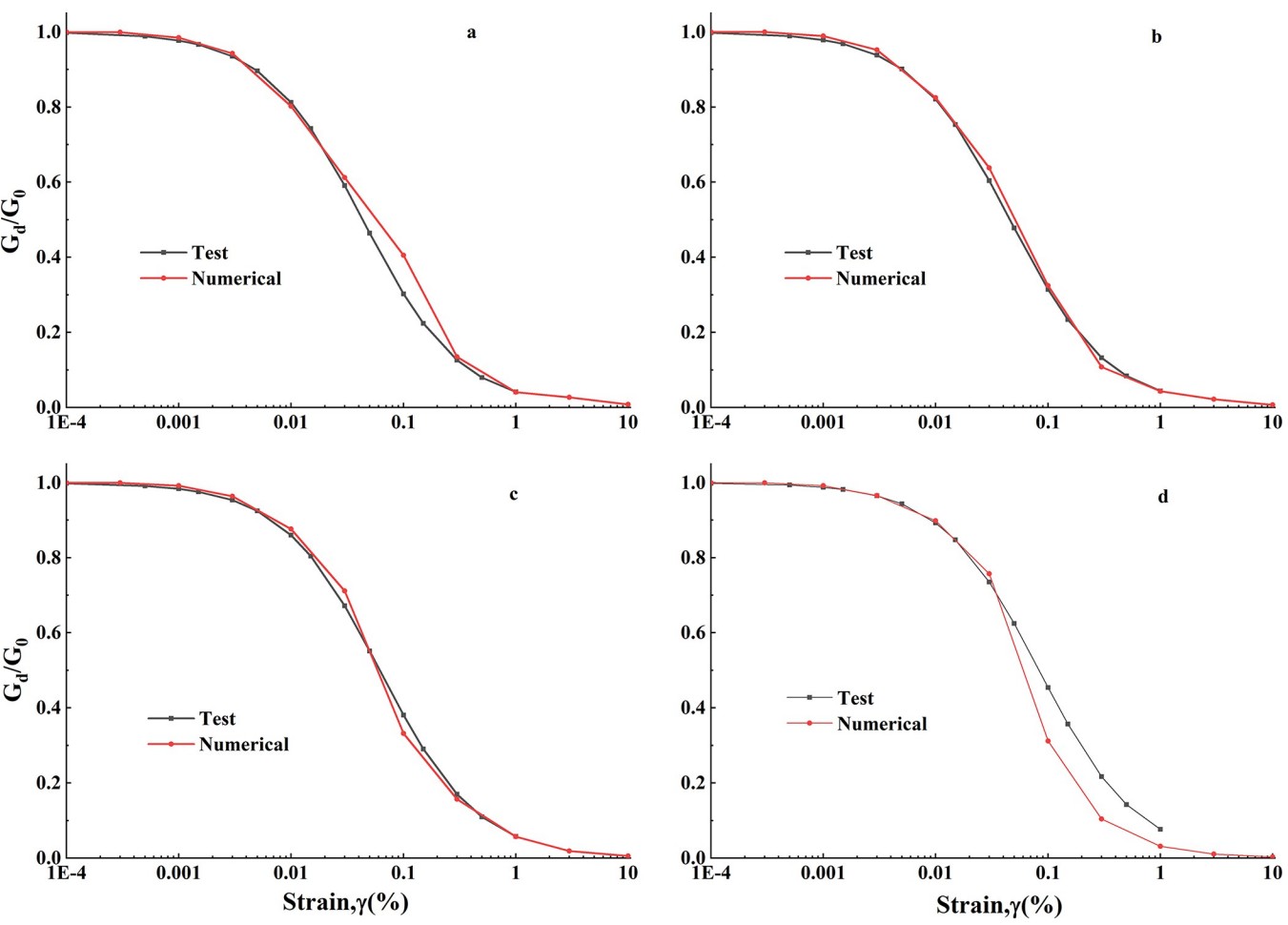

**Fig 5. $G_d/G_0$-$\gamma$ Chart of test and numerical results.** a. layer1; b. layer2; c. layer3; d. layer4.

follows:

$$\varepsilon = \frac{A_{(2,3...i-1)}}{A_1} \qquad (3)$$

Here, $A_{(2,3...i-1)}$ represents the peak ground acceleration on the loess ridges surface, and $A_1$ represents the peak ground acceleration on the loess ridges surface at the slope toe.

## 3.1 low-intensity seismic motion response

As shown in the Fig 6A, the point-line graph illustrates the $\varepsilon$-$\eta$ relationship on the slope surface for an input seismic wave with a peak acceleration of 0.1 m/s$^2$ and a frequency of 1 Hz. It can be observed that when the slope is relatively low (50m, 60m), $\varepsilon$ consistently exceeds 1 and increases progressively with higher $\eta$ values. When $\eta$ equals 1, which corresponds to the point of transition from a linear slope to a curved slope top (defined here as the slope shoulder) for the H = 50m model, $\varepsilon$ reaches its maximum value at 1.59, while the H = 60m model at the same location yields an $\varepsilon$ of 1.32. For slopes with heights between 70m and 100m, $\varepsilon$ exhibits a trend where it first decreases and then increases as $\eta$ increases. Specifically, when $\eta$ falls within the range of 0–0.5, $\varepsilon$ remains less than 1, decreasing as $\eta$ increases. However, when $\eta$ falls

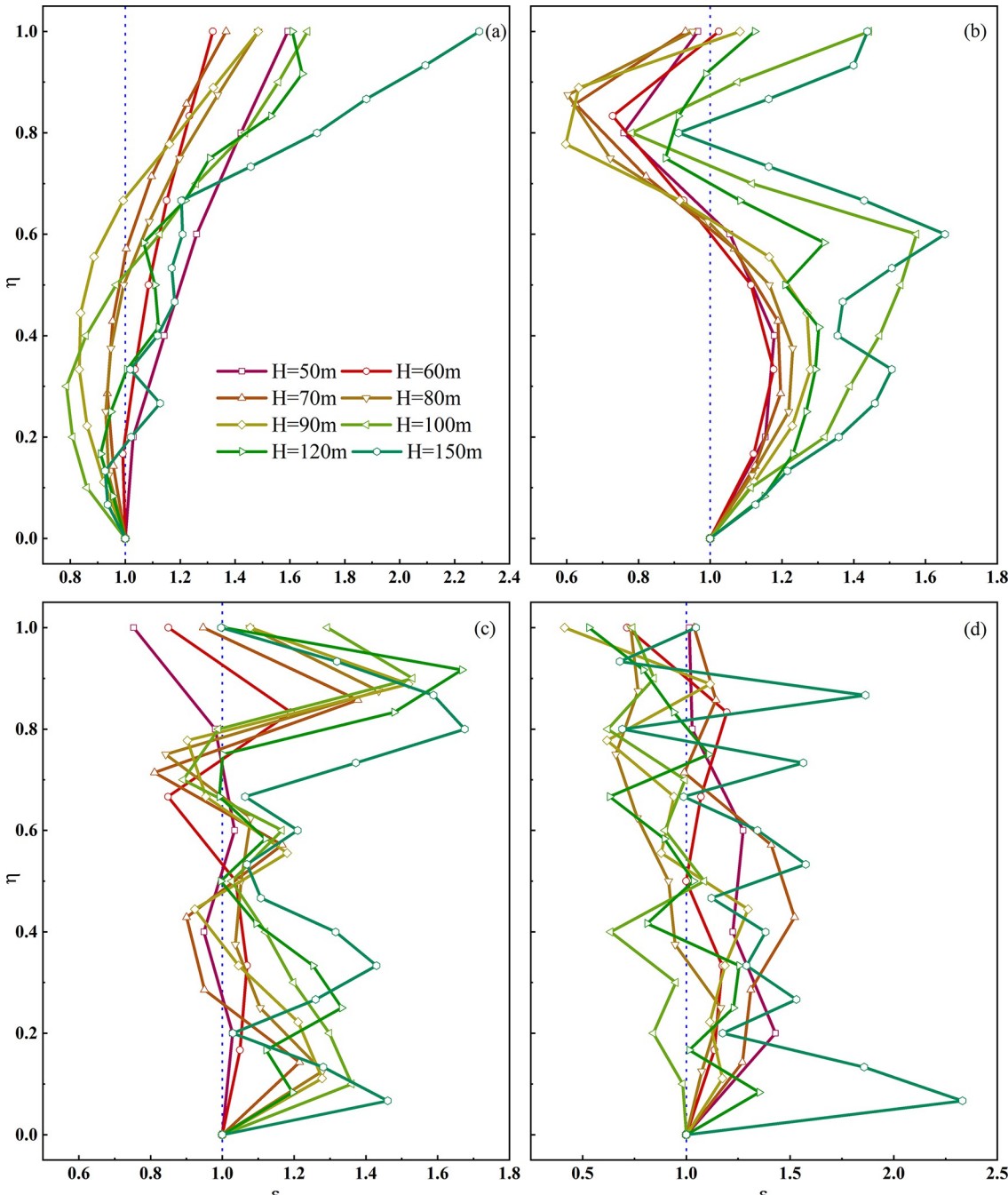

**Fig 6. ε-η curves on the slope surface of loess ridge at different frequencies (*f*).** a. *f* = 1Hz, b. *f* = 2Hz, c. *f* = 5Hz, d. *f* = 10Hz.

within the range of 0.5–1, ε increases with η. For η values greater than 0.7, ε consistently exceeds 1. At the slope shoulder, where η equals 1, ε increases with higher slope heights. For instance, when H = 70m, ε equals 1.37, and when H = 100m, ε equals 1.66. When the slope height is 120m or 150m, ε, as a whole, increases with higher η values but exhibits a regular undulating pattern: starting from η = 0, ε shows a rhythmic change of decreasing, increasing, decreasing, and increasing. When η equals 1, ε reaches its maximum, with values of 1.61 and 2.29 for the H = 120m and H = 150m models, respectively. For low and short loess ridges

(50m, 60m), as the elevation increases, the acceleration steadily increases. In the case of moderate slopes (70m-100m), when η is less than 0.5, ε decreases, but it increases when η exceeds 0.5 times the slope height. In the context of tall ridges (120m, 150m), the acceleration, on the whole, shows an increasing trend. However, it exhibits irregular fluctuations of decreasing and increasing.

As shown in Fig 6B, at a frequency of 2Hz, ε exhibits a regular pattern of increase-decrease-increase with increasing η. Specifically, when η is in the range of 0–0.3, ε is consistently greater than 1 and increases with higher η values. Within this range, for larger H values, ε becomes larger. For instance, at H = 50m, ε is 1.18, and at H = 150m, ε is 1.51.

When η falls within the range of 0.3–0.8, ε decreases with increasing η, and at η = 0.8, all models have ε values less than 1. The model with H = 90m has the smallest ε in this interval, equivalent to only 0.63 times the value at the slope toe. Subsequently, ε increases rapidly with increasing η, reaching around 1 near the slope shoulder.

At 2Hz, alternating peaks and troughs in ε appear on the loess double-sided slope, and as the slope height increases, the difference between the maximum and minimum values of ε becomes larger.

When subjected to seismic motion with a frequency of 5Hz, ε exhibits a regularly spaced, alternated distribution of maximum and minimum values as η increases, showing a rhythmic pattern of alternation between amplification and reduction. However, the average of these alternating amplifications and reductions increases with the increase in slope height. At 5Hz, as shown in Fig 6C, the average values for H = 50m, 60m, 70m, 80m, 90m, 100m, 120m, and 150m are 0.95, 1.00, 1.05, 1.11, 1.12, 1.15, 1.17, and 1.28, respectively. At 10Hz, As shown in Fig 6D, the frequency at which maximum and minimum values alternate is faster than at 5Hz. The average ε values for H = 50m, 60m, 70m, 80m, 90m, 100m, 120m, and 150m are 1.19, 1.04, 1.24, 0.88, 0.97, 0.86, 0.97, and 1.36, respectively.

In summary, under seismic action on the loess double-sided slope, ε exhibits significant nonlinear characteristics. The rhythmic alternation between amplification and reduction of ε becomes more pronounced with increasing slope height and seismic motion frequency.

The seismic response at the crest of the slope is illustrated in Fig 7. It represents the ratio of seismic motion at the crest to that at the slope toe. Observations from the graph reveal that, under 1Hz seismic motion, the larger the height ($H_{max}$) of the models at the crest, the greater the ε. For instance, when $H_{max}$ = 60m, ε is 1.42, and when $H_{max}$ = 150m, ε is 1.93. ε exhibits a linear relationship with $H_{max}$, increasing by 0.05 for every 10 m of crest height. Under 2Hz seismic motion, all models have ε values lower than those at 1Hz, and the ε- $H_{max}$ relationship also follows a linear trend, with a slope consistent with the 1Hz condition. At 5Hz, ε for models with $H_{max}$ between 60m and 100m is lower than 1, and it increases linearly with $H_{max}$, following a trend similar to the one discussed above. Under 10Hz seismic motion, ε exceeds 1 only at $H_{max}$ = 160m; for other heights, it remains below 1.

The amplification factor at the crest of the loess double-sided slope decreases as the frequency increases. At lower frequencies, the amplification factor linearly increases with the crest height, while at higher frequencies, the effect of crest height is less pronounced.

### 3.2 Strong-intensity seismic motion response

Under the influence of strong seismic activity, the soil experiences larger deformations and strains, causing changes in its dynamic characteristics, and subsequently altering the acceleration amplification effects. In a uniform soil layer scenario, when subjected to higher seismic intensities, the soil undergoes increased shear strain, leading to a decrease in shear modulus and an increase in damping ratio [37]. Consequently, the fundamental frequency of the site

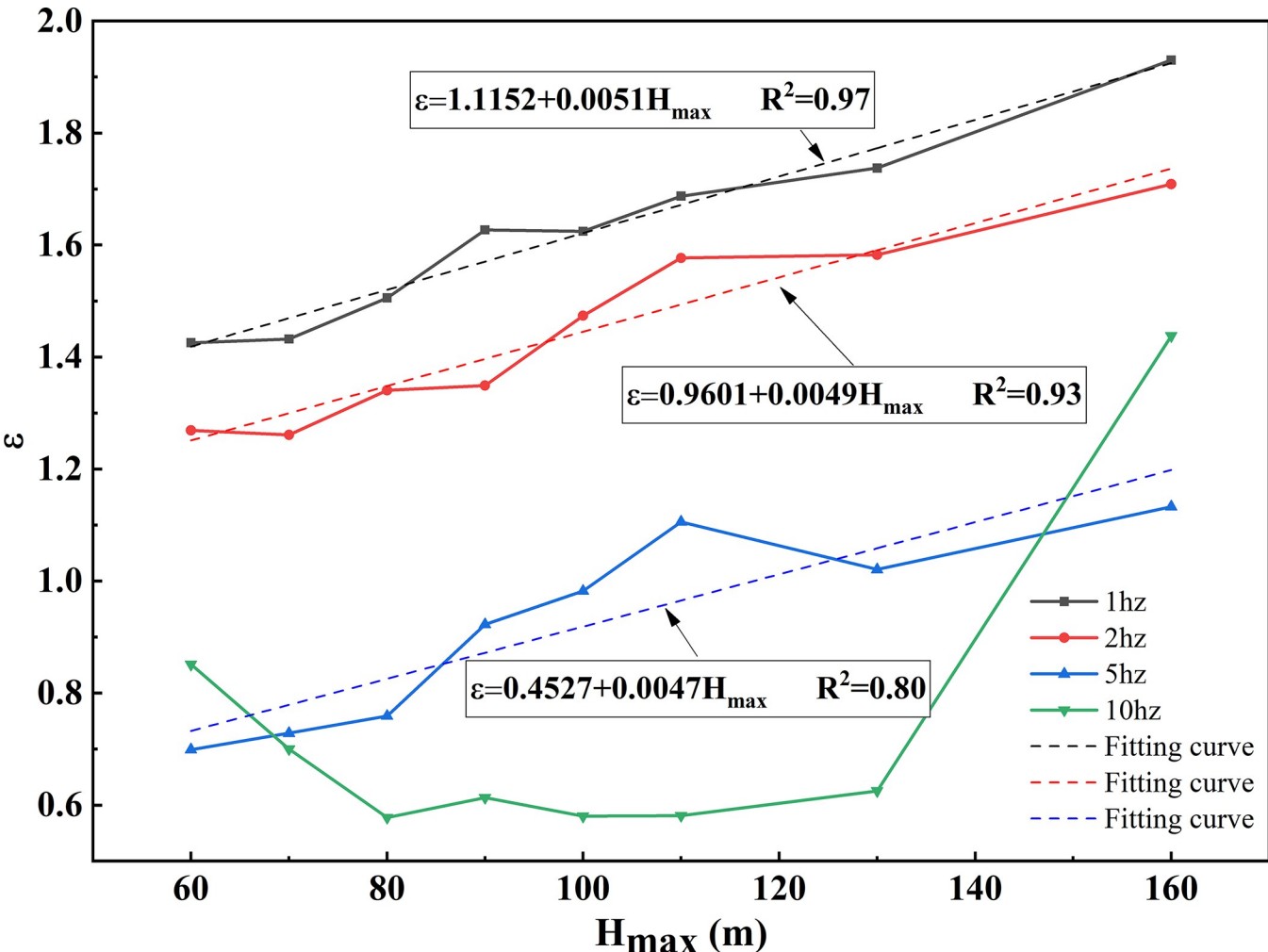

**Fig 7. ε-H$_{max}$ curves on the crest surface of loess ridge at different frequencies.**

decreases, along with a shift in the amplification frequency range, while the increased damping ratio contributes to higher energy dissipation for high-frequency seismic motions.

As illustrated in Fig 8, the chart depicts the ε values at different locations for models subjected to seismic accelerations with an input of 1 m/s². In comparison to small seismic motions (0.1 m/s²), the ε-η curves under 1 Hz seismic activity exhibit a generally consistent growth pattern across various models. However, the significant difference lies in the fact that under strong seismic conditions, ε values tend to be higher in most models, and the ε-η curves transition from a smoother pattern observed during small seismic motions to a more intricate and irregular behavior.

For seismic vibrations with a frequency of 2 Hz, the variation patterns are essentially akin to those observed at 1 Hz. When exposed to 5 Hz and 10 Hz seismic forces, especially in the case of the low and short loess ridges (H = 50m, 60m), a pronounced rhythmic pattern becomes more apparent. Additionally, for slopes with greater heights, the dispersion of ε values between maximum and minimum values decreases.

As shown in Fig 9, under strong seismic activity, the ε values at the crest of the loess double-sided slope show the following ranges:

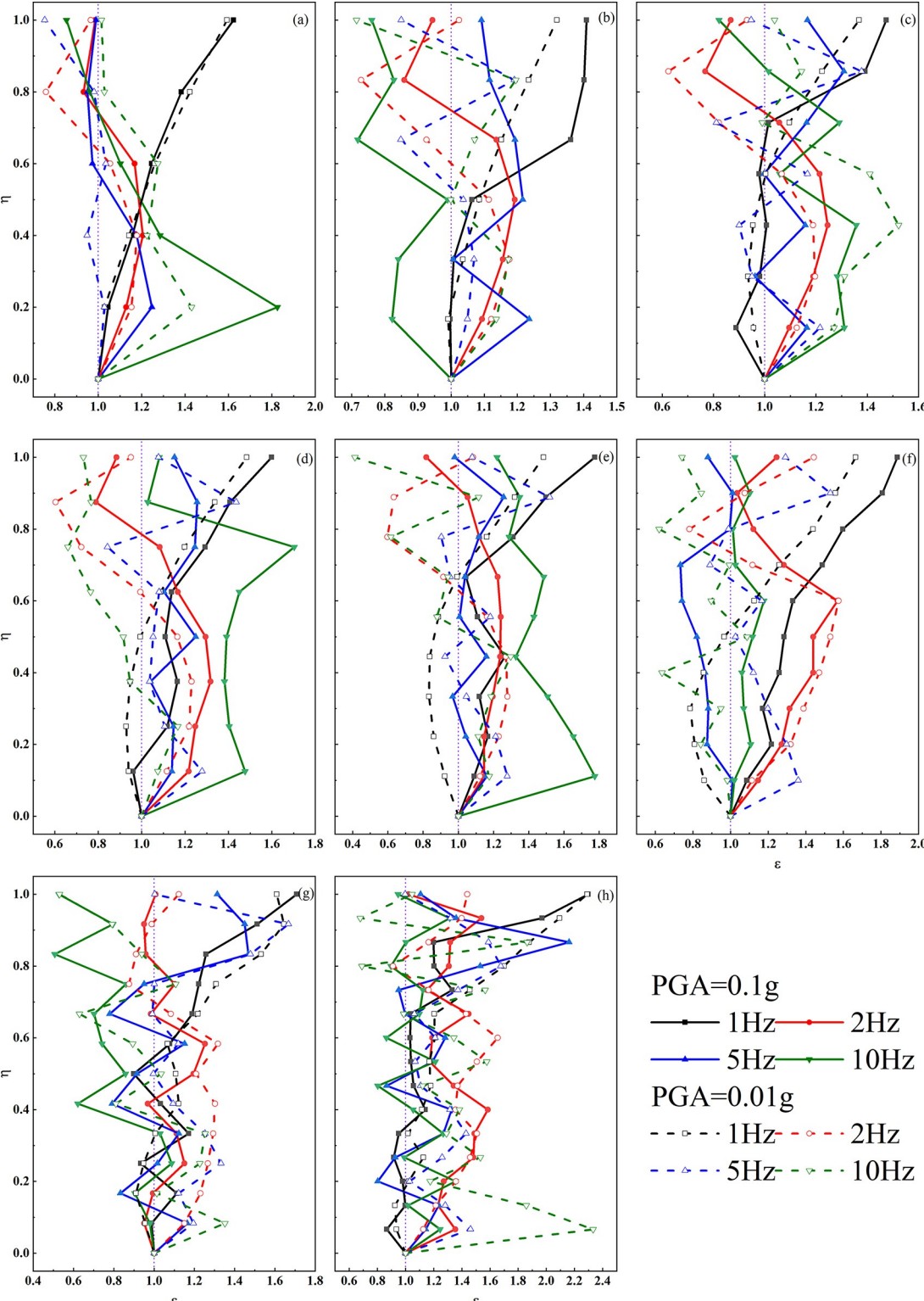

**Fig 8. ε-η curves of loess ridges at different heights under high-intensity earthquake motions.** a. H = 50m; b. H = 60m; c. H = 70m; d. H = 80m; e. H = 90m; f. H = 100m; g. H = 120m; h. H = 150m.

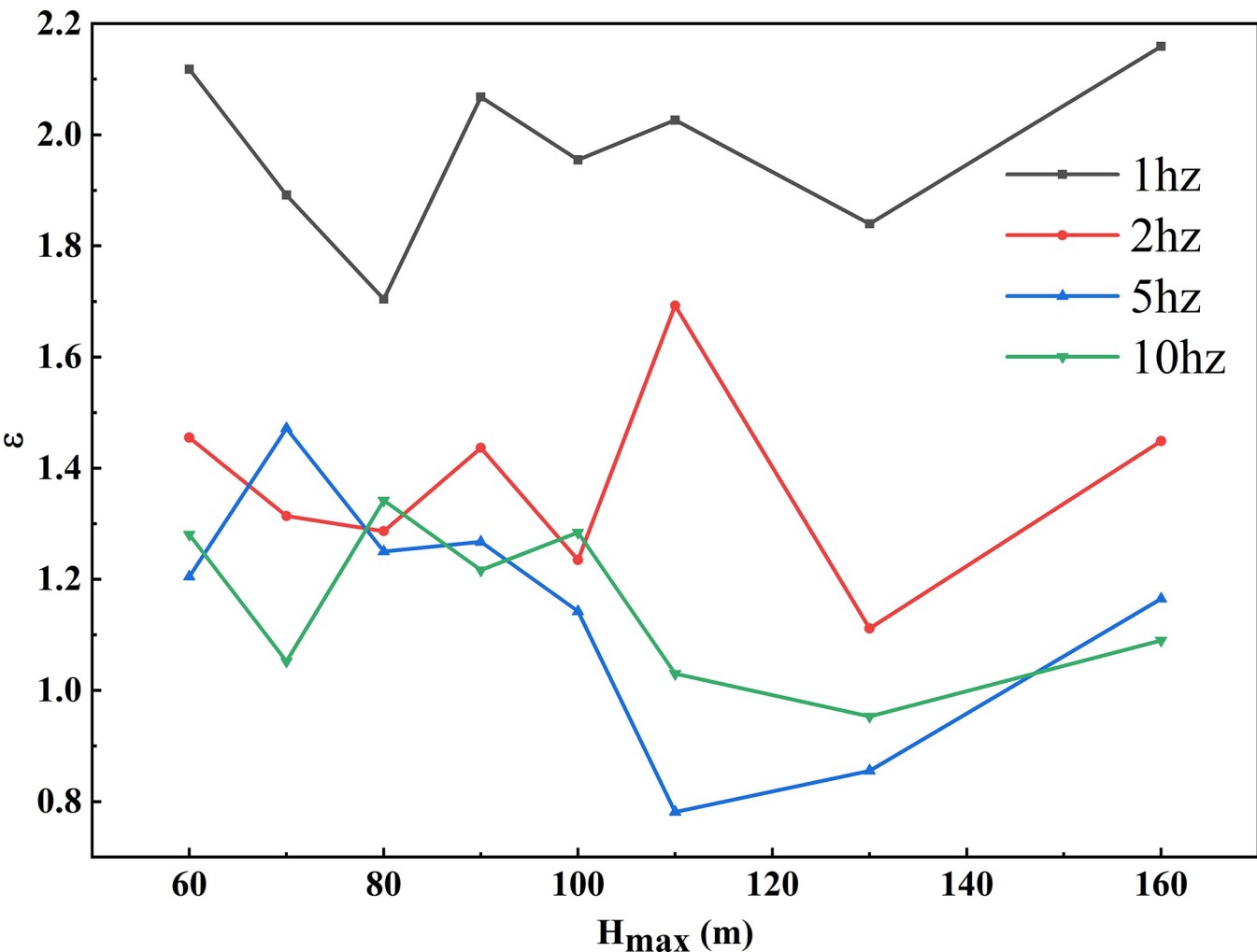

**Fig 9. ε-H$_{max}$ curves on the crest surface of loess ridge at different frequencies under high-intensity motion.**

- At 1 Hz: Ranging from 1.70 to 2.16, with an average of 1.99.

- At 2 Hz: Ranging from 1.11 to 1.69, with an average of 1.37.

- At 5 Hz: Ranging from 0.78 to 1.47, with an average of 1.18.

- At 10 Hz: Ranging from 0.95 to 1.34, with an average of 1.15.

Notably, the shared observation is that ε decreases with an increase in frequency. Nevertheless, there are two distinctive differences: first, ε values, on the whole, are greater, significantly surpassing ε under small seismic motions; second, the linear trend of ε with respect to slope height disappears, and ε no longer increases with greater slope height.

The previous analysis employed single-frequency artificial time history records to study the impact of seismic intensity and frequency on the terrain effects of loess ridges. However, natural seismic waves are characterized by stochastic signals containing various frequency components. To address this, we utilized acceleration time histories recorded at the Minxian Station, located 18 km from the epicenter of an $M_s$ 6.6 earthquake that occurred in loess areas. This natural seismic record, with a predominant frequency around 4 Hz, was compared to the 1 m/s$^2$ Chang's signals at 2 Hz and 5 Hz. Fig 10 (10a-10h) depicted the ε-η curves for models with

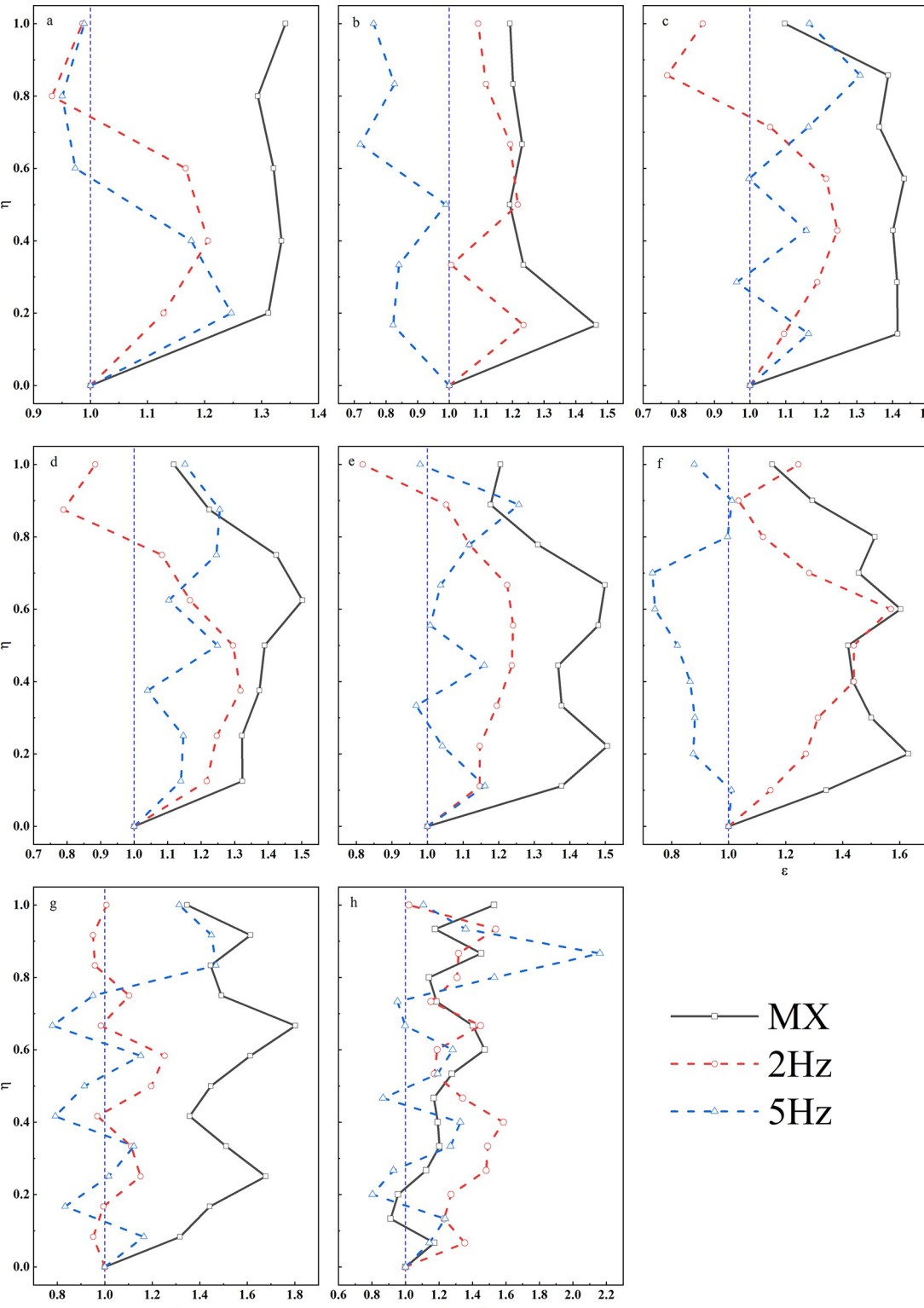

**Fig 10. ε-η curves of loess ridges at different heights under Minxian earthquake wave.** a. H = 50m; b. H = 60m; c. H = 70m; d. H = 80m, e. H = 90m, f. H = 100m, g. H = 120m, h. H = 150m.

slope heights (H) ranging from 50 to 150 m. In general, the ε-η curve for the Min County seismic record exhibited a changing trend similar to that of the 2 Hz signal, displaying relatively consistent periodic variations. However, at the same locations, the ε values for the slope with H = 60m were noticeably higher than those obtained using the Chang's 2 Hz signal.

The amplification results at the crest show that when $H_{max}$ = 90m, the ε value at the crest is minimized at 1.45, and when $H_{max}$ = 120m, the ε value at the crest is maximized at 2.14. This result is less than the ε value at a frequency of 1Hz with a peak acceleration of 0.1g and greater than the ε value at a frequency of 2Hz with a peak acceleration of 0.1g (Fig 11). Similar to the research findings of Paolucci [38] on double-sided slopes, the relative amplification at the crest of loess ridges is in the range of 1.5 to 2 times.

The Fourier spectral ratios of three points at the slope shoulder and crest positions relative to the slope toe position were selected, and the results are shown in the Fig 12 and Table 2. In the low-frequency range (0.1-1Hz), there is a significant amplification effect at points Ai and Ai-1, while the amplification effect at A2 is less pronounced. The ε value at point Ai is greater than that at Ai-1, and A2 has the lowest ε value. Within this interval, there exists a narrow range of maximum ε values, spanning from 0.1 to 0.3Hz. The maximum ε value for Ai can reach up to 10.59, while Ac reaches 5.42. The ε values generally fall within the range of 2.5 to 4.

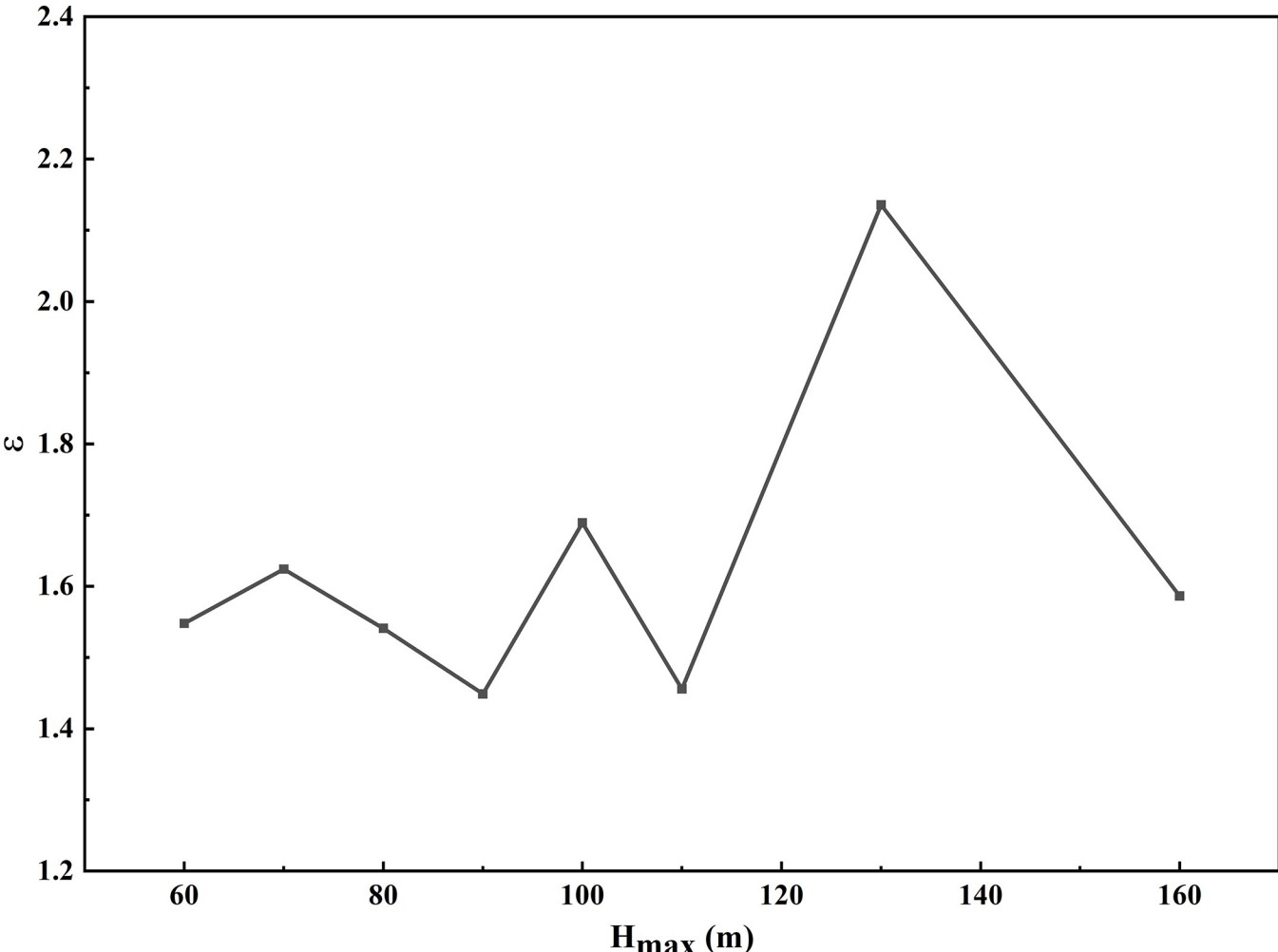

**Fig 11. ε-$H_{max}$ curves on the crest surface of loess ridge under Minxian earthquake wave.**

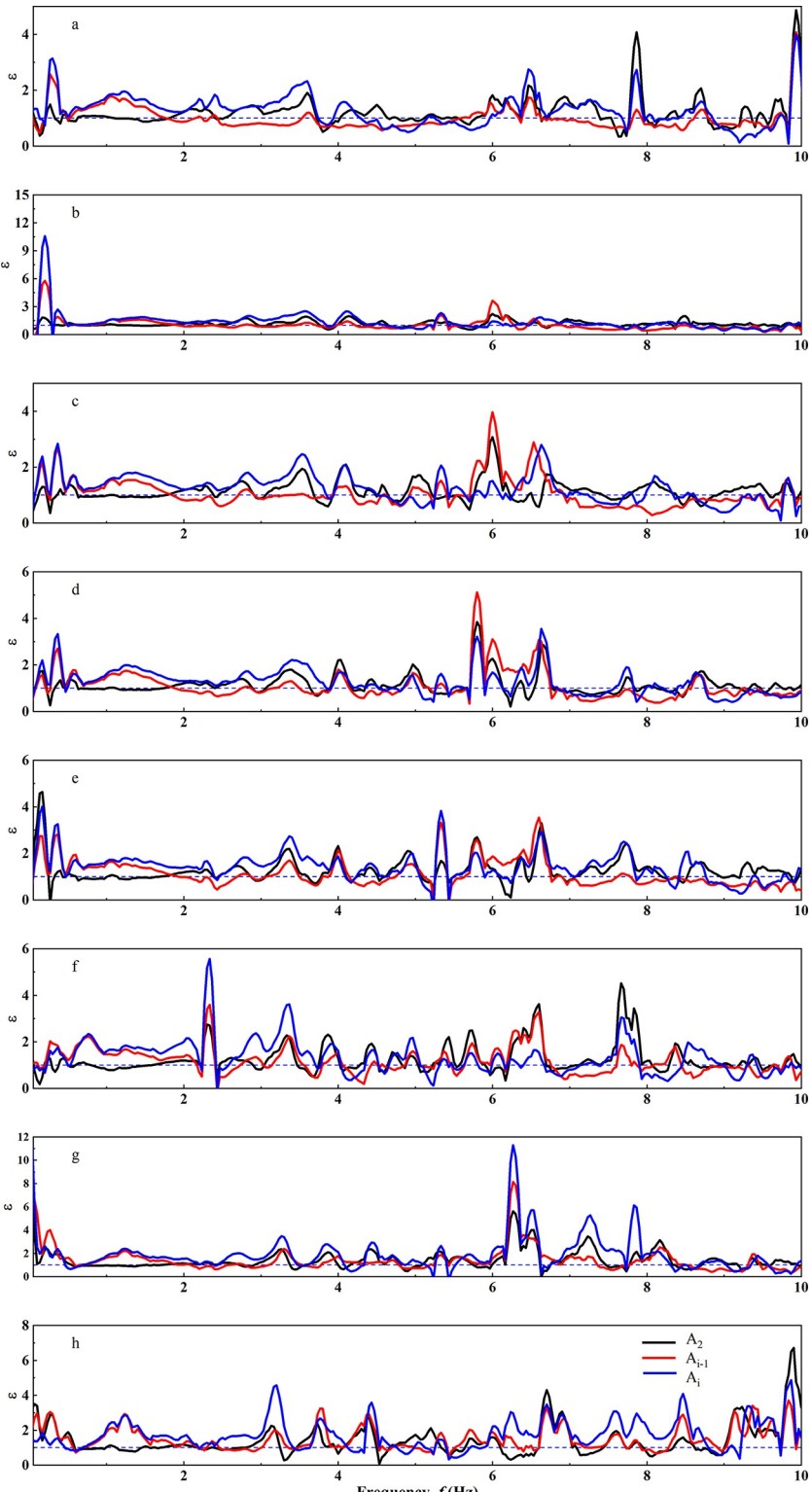

**Fig 12. Fourier spectrum ratio curve.** under Minxian earthquake wave. a. H = 50m; b. H = 60m; c. H = 70m; d. H = 80m; e. H = 90m; f. H = 100m; g. H = 120m; h. H = 150m.

**Table 2. Maximum amplification effects and corresponding frequencies for different frequency bands of seismic motions.**

| H(m) | Maximum amplification factors (AF) and corresponding frequencies (f) | | | | | | | | | | | | | | | | | |
|---|---|---|---|---|---|---|---|---|---|---|---|---|---|---|---|---|---|---|
| | 0.1-1Hz | | | | | | 1-5Hz | | | | | | 5-10Hz | | | | | |
| | $A_2$ | | $A_{i-1}$ | | $A_i$ | | $A_2$ | | $A_{i-1}$ | | $A_i$ | | $A_2$ | | $A_{i-1}$ | | $A_i$ | |
| | AF | f | AF | f | AF | f | AF | f | AF | f | AF | f | AF | f | AF | f | AF | f |
| 50 | 1.50 | 0.27 | 2.57 | 0.27 | 3.14 | 0.30 | 1.90 | 3.60 | 1.78 | 1.07 | 2.32 | 3.60 | 4.86 | 9.93 | 4.08 | 9.93 | 4.00 | 9.93 |
| 60 | 1.83 | 0.17 | 5.42 | 0.17 | 10.59 | 0.20 | 1.97 | 4.13 | 1.63 | 1.06 | 3.57 | 2.50 | 2.22 | 6.00 | 3.62 | 6.00 | 2.30 | 5.33 |
| 70 | 1.37 | 0.53 | 2.69 | 0.37 | 2.84 | 0.37 | 2.09 | 4.10 | 1.55 | 1.07 | 2.47 | 3.53 | 3.08 | 6.00 | 3.97 | 6.00 | 2.79 | 6.63 |
| 80 | 1.74 | 0.17 | 2.70 | 0.37 | 3.33 | 0.37 | 2.22 | 4.03 | 1.78 | 4.00 | 2.22 | 3.40 | 3.84 | 5.80 | 3.80 | 5.12 | 3.55 | 6.30 |
| 90 | 4.64 | 0.17 | 2.81 | 0.37 | 4.02 | 0.17 | 2.32 | 4.00 | 2.14 | 4.00 | 2.74 | 3.37 | 3.30 | 6.63 | 3.53 | 6.60 | 3.82 | 5.33 |
| 100 | 1.33 | 0.23 | 2.23 | 0.77 | 2.23 | 0.77 | 2.74 | 2.30 | 3.59 | 2.33 | 5.56 | 2.33 | 4.52 | 7.67 | 3.27 | 6.60 | 3.05 | 7.67 |
| 120 | 2.36 | 0.23 | 4.01 | 0.27 | 2.61 | 0.20 | 2.36 | 3.26 | 2.37 | 3.30 | 3.47 | 2.37 | 5.61 | 6.27 | 8.13 | 6.27 | 11.29 | 6.27 |
| 150 | 3.39 | 0.10 | 3.03 | 0.27 | 1.87 | 0.17 | 2.71 | 4.40 | 3.24 | 3.77 | 4.57 | 3.20 | 6.71 | 9.90 | 3.70 | 9.83 | 4.87 | 9.87 |

The high-frequency range of 1-10Hz is divided into two parts: 1-5Hz and 5-10Hz. Ai exhibits significant amplification from 1 to 4Hz, with a narrow range of maximum amplification occurring between 3.2–3.6Hz, ranging from 2.22 to 2.74. Amplification is less pronounced from 4-5Hz. $A_h$ shows notable amplification from 1 to 2Hz, while other segments exhibit less amplification. In the 5-10Hz range, A2's ε values are greater than those of Ai and Ai-1. Within this interval, there are several narrow ranges of maximum ε values around 6.3Hz and 10Hz, with the maximum ε value reaching 11.29.

The amplification effect at the crest position of the site does not increase with the increasing slope height. The ε values, as the slope height increases, show more frequent occurrences of narrow-range maximum values, resulting in greater amplification at high frequencies. This is related to the fact that higher models have higher natural frequencies.

## 4. Conclusions

Terrain effects are important considerations for site selection and seismic design of buildings to avoid or mitigate the potential risk of earthquake disasters. This study aimed to obtain the characteristics of acceleration amplification on the slopes of loess ridges under seismic actions by establishing a generalized numerical model of loess ridges. The analysis of seismic effects on loess ridges led to the following conclusions:

1. The seismic amplification on the slopes of loess ridges does not exhibit a linear increase with slope height. Instead, it demonstrates a rhythmic pattern of increase and decrease, with the rate of change in this pattern increasing with seismic frequency and slope height.

2. The seismic amplification at the crest of loess ridges decreases with increasing seismic frequency and increases with higher input seismic intensity.

3. Under low-intensity seismic conditions, the seismic amplification at the crest of loess ridges linearly increases with slope height. However, under high-intensity seismic conditions, the relationship between amplification and slope height is less pronounced, with peak accelerations at the crest typically being 1.5 to 2.5 times greater than those at the slope toe.

4. The amplification effect at the crest of loess ridges is most significant in the low-frequency and high-frequency portions, with remarkable amplification effects near the base of the slope in the high-frequency range. Additionally, the analysis reveals the presence of several extremely narrow frequency bands with maximum amplification, where the amplification factor can reach over tenfold.

This study provides insights into the terrain amplification effects on loess ridges based on numerical simulations, offering valuable guidance for engineering construction and seismic disaster reduction planning in loess regions. Given the complexity of actual terrain, the diversity of soil parameters, and the uncertainty of seismic motion, the idealized model presented in this paper should be compared with observed seismic results to obtain a more reasonable understanding of the terrain effects on loess ridges.

## Supporting information

**S1 Data.**
(XLSX)

## Author Contributions

**Data curation:** Da Peng.

**Formal analysis:** Da Peng.

**Funding acquisition:** Jingshan Bo.

**Investigation:** Da Peng.

**Methodology:** Da Peng, Wenhao Qi.

**Project administration:** Jingshan Bo.

**Software:** Da Peng.

**Validation:** Xiaobo Li.

**Writing – original draft:** Da Peng.

**Writing – review & editing:** Da Peng, Chaoyu Chang.

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
