## [Decision Letter · Decision Letter 0]

21 Nov 2023

PONE-D-23-36127Numerical Investigation of Seismic Amplification Characteristics in Loess RidgePLOS ONE

Dear Dr. Peng,

Thank you for submitting your manuscript to PLOS ONE. After careful consideration, we feel that it has merit but does not fully meet PLOS ONE’s publication criteria as it currently stands. Therefore, we invite you to submit a revised version of the manuscript that addresses the points raised during the review process.

We look forward to receiving your revised manuscript.

Kind regards,

Dr. S. M. Anas, Ph.D.(Structural Engg.), M.Tech(Earthquake Engg.)

Academic Editor

PLOS ONE

Journal Requirements:

3. Thank you for stating the following financial disclosure: "This work is financially supported by the National Natural Science Foundation of China (No. U1939209),Scientific Research Fund of Institute of Engineering Mechanics, China Earthquake Administra-tion (Grant NO.2020EEEVL0201)

4. Thank you for stating the following in your Competing Interests section: "NO authors have competing interests".

Additional Editor Comments:

Dear Authors:

The revised manuscript entitled "Numerical Investigation of Seismic Amplification Characteristics in Loess Ridge" [PONE-D-23-36127] was sent to the two peer-reviewers who were experts in the concerned research area: one suggested "Minor Revision" and the other recommended for "Major Revision" and gave very serious comments about the quality, content, novelty and layout of the manuscript. Based on the reviewers' recommendations and preliminary analysis of the manuscript, this editor has decided to take "Major Revision" decision on the above submission and suggest authors carefully address the issues/concerns raised by the reviewers.

Thank you very much for submitting your manuscript to PLOS ONE.

We look forward to receiving your revised version of the manuscript.

Sincerely yours,

Dr. S. M. Anas

(Academic Editor)

Reviewers' comments:

Reviewer's Responses to Questions

**Comments to the Author**

1. Is the manuscript technically sound, and do the data support the conclusions?

Reviewer #1: Yes

Reviewer #2: Yes

2. Has the statistical analysis been performed appropriately and rigorously? 

Reviewer #1: Yes

Reviewer #2: N/A

3. Have the authors made all data underlying the findings in their manuscript fully available?

Reviewer #1: No

Reviewer #2: No

4. Is the manuscript presented in an intelligible fashion and written in standard English?

Reviewer #1: Yes

Reviewer #2: Yes

5. Review Comments to the Author

Reviewer #1: The research article focuses on the seismic response of loess double-sided slopes. The study employs numerical modeling to investigate the effects of seismic actions on the slopes of loess ridges. The study provides insights into terrain effects under seismic conditions and offers guidance for construction and disaster planning in loess regions.

Please take the following comments and suggestions into account and provide a revised, improved manuscript based on them:

1. in page 4, please provide more information about the Loess Plateau region. Where it is, what it is, etc, as this information is not known to the reader. Probably this is well known in China, but the target audience of the journal is worldwide so this needs to be explained further and in detail

2. a general comment is about the generalization of the results: How applicable are the findings to other loess regions with varying soil properties?

How can the study's findings be generalized to other geographic regions with loess formations? Are there specific soil properties or environmental conditions where these findings might not be applicable?

3. Frequency Range Analysis: You mention that the frequency (f) of the seismic input time history is taken at 1 Hz, 2 Hz, 5 Hz, and 10 Hz. Is there a specific reason for choosing the specific frequency range used in the study? What was the scientific basis for choosing the specific frequency range used in this study? How do different frequency ranges influence the seismic amplification observed in loess slopes?

4. The authors could elaborate more on how each variable within the numerical model contributes to the overall seismic response. Specifically, understanding the individual impact of variables like soil density, cohesion, and internal friction angle would be beneficial. Additionally, how were these variables calibrated or adjusted to reflect real-world conditions more accurately?

5. How does the temporal aspect of seismic activity (e.g., duration and frequency of seismic events over time) influence the long-term stability of loess slopes? It would be insightful if the study included a temporal analysis to predict changes in slope stability over extended periods of seismic activity.

6. Considering the complexity of soil-structure interactions during seismic events, how might the presence of buildings or other structures on or near the loess slopes impact the findings? A deeper exploration into how structures modify the seismic wave propagation and amplification would add value to the study.

7. The integration of historical seismic data into the numerical model could enhance the study's relevance. How might the incorporation of past seismic records from the study area refine the model's predictions? This approach could validate the model's accuracy in simulating real-world scenarios.

8. Loess soil properties can significantly change with varying water content. How would the presence of underground water or a varying water table affect the seismic response of loess slopes? A discussion on the impact of hydrological conditions would provide a more comprehensive understanding of the seismic behavior of loess soils.

9. Apart from seismic activities, loess slopes are also subjected to non-seismic ground movements such as landslides or erosion. How do these factors interplay with seismic forces in influencing slope stability? An analysis of the combined effects would provide a more holistic view of the risks associated with loess slopes.

10. A sensitivity analysis of key parameters (e.g., slope height, soil properties, seismic intensity) could provide insights into their relative importance in the model. Which parameters are most sensitive, and how do changes in these parameters affect the overall seismic response?

11. How would the seismic response of loess slopes compare with slopes made of different soil types, such as clay or sandy soils? A comparative study could highlight the unique aspects of loess soil behavior under seismic stress.

12. Based on the findings of the study, what practical guidelines or recommendations can the authors provide for engineering applications, such as construction practices, slope reinforcement techniques, and early warning systems? Detailed suggestions would make the study more applicable to engineers and urban planners working in loess regions. How can the findings of this study be translated into practical applications for disaster planning and risk mitigation in regions with loess formations? Are there specific strategies or tools that the authors would recommend based on their research?

13. The authors are requested to provide a comprehensive explanation of the criteria and rationale behind the selection of the specific numerical models used in this study. How do these models compare with other available models in terms of accuracy, reliability, and relevance to the seismic behavior of loess slopes?

14. Have the results of the numerical models been validated against actual seismic event data? If so, how closely do the model predictions align with real-world observations, and what discrepancies were noted?

15. What are the acknowledged limitations of the generalized numerical model applied in this study? How might these limitations impact the interpretation and application of the findings? Please include these and the relevant discussion in the revised manuscript.

16. How does the seismic response of loess slopes vary with different levels of seismic intensity? Are there specific intensities where the amplification effects become significantly more pronounced?

17. To what extent do variations in soil parameters such as moisture content, density, and composition affect the conclusions drawn from this study? Could the authors elaborate on how these factors were accounted for in their analysis?

18. How do factors of uncertainty in seismic motion (such as unpredictable intensity and duration) affect the conclusions of the study? Were these uncertainties considered in the modeling process?

19. What areas for future research do the authors identify based on their findings? Are there specific aspects of loess slope seismic behavior that require further investigation?

20. please make sure you use proper superscripts, etc in units and others, for example in table 1 (kg/m3). "3" should be a superscript.

21. fonts are not properly scaled in figures. see for example fig 4 where the fonts used for the axes of the graph are very small and can hardly be read. the same for fig 5 and others. fonts should be properly scaled. in fact most figures have this problem, I think. Please update ALL figures with proper fonts that can be read either on the screen or on the paper. the same for table 2 where again fonts are too small, but this can be fixed by the journal later on.

Reviewer #2: Ref paper: PONE-D-23-36127

The paper entitled " Numerical Investigation of Seismic Amplification Characteristics in Loess Ridge ", by Da Peng, Jingshan Bo, Chaoyu Chang,Wenhao Qi and Xiaobo Li is a work investigating ways, to determine procedures quantifying the peak acceleration amplification effects experienced by the surface of loess ridges when subjected to SV waves In this study, generalized geometric models of loess ridges were created, and a three-dimensional nonlinear numerical model was established using FLAC3D, considering different seismic input ground motion time histories.

Some very interesting conclusions are derived which could be applied for engineering construction and seismic risk mitigation planning in loess regions. A great number of constructive figures are included in the manuscript.

I recommend this paper for publication in PLOS ONE

In order to improve the final version of the paper:

1- I suggest to the authors to make a comparison of their results obtained in this study, with previous similar works, as a validation or verification.

2-The paper is interesting, however, requires re-writing in terms of technical English for better understanding of wider audience ship before publication.

6. PLOS authors have the option to publish the peer review history of their article (what does this mean?). If published, this will include your full peer review and any attached files.

Reviewer #1: No

Reviewer #2: No

---

## [Author Response · Author response to Decision Letter 0]

19 Dec 2023

We would like to thank you for your careful reading, helpful comments, and constructive suggestions, which has significantly improved the presentation of our manuscript.

We have carefully considered all comments from the reviewers and revised our manuscript accordingly. The manuscript has also been double-checked, and the typos and grammar errors we found have been corrected. In the following section, we summarize our responses to each comment from the reviewers. We believe that our responses have well addressed all concerns from the reviewers. Many thanks to the reviewer for providing valuable feedback, especially for guiding us in our future research directions.

We hope our revised manuscript can be accepted for publication. The response as shown in attachment.

---

## [Decision Letter · Decision Letter 1]

2 Jan 2024

Numerical Investigation of Seismic Amplification Characteristics in Loess Ridge Region of Xiji, Northwest China

PONE-D-23-36127R1

Dear Dr. Peng,

We’re pleased to inform you that your manuscript has been judged scientifically suitable for publication and will be formally accepted for publication once it meets all outstanding technical requirements.

Kind regards,

Dr. S. M. Anas, Ph.D.(Structural Engg.), M.Tech(Earthquake Engg.)

Academic Editor

PLOS ONE

Additional Editor Comments (optional):

Esteemed Corresponding Author and Co-Authors,

The revised manuscript, titled "Numerical Investigation of Seismic Amplification Characteristics in Loess Ridge Region of Xiji, Northwest China" [PONE-D-23-36127R1], has undergone a reevaluation by the previous reviewers. I am pleased to inform you that the reviewers have accepted the revised manuscript in its current form. Based on their recommendations, I have decided to proceed with an "Accept" decision for this version, pending approval from the editorial board.

We appreciate your submission to PLOS ONE and anticipate receiving your future contributions.

Best regards,

Dr. S. M. Anas

Academic Editor

PLOS ONE

Reviewers' comments:

Reviewer's Responses to Questions

**Comments to the Author**

1. If the authors have adequately addressed your comments raised in a previous round of review and you feel that this manuscript is now acceptable for publication, you may indicate that here to bypass the “Comments to the Author” section, enter your conflict of interest statement in the “Confidential to Editor” section, and submit your "Accept" recommendation.

Reviewer #1: All comments have been addressed

Reviewer #2: All comments have been addressed

2. Is the manuscript technically sound, and do the data support the conclusions?

Reviewer #1: Yes

Reviewer #2: (No Response)

3. Has the statistical analysis been performed appropriately and rigorously? 

Reviewer #1: Yes

Reviewer #2: N/A

4. Have the authors made all data underlying the findings in their manuscript fully available?

Reviewer #1: Yes

Reviewer #2: Yes

5. Is the manuscript presented in an intelligible fashion and written in standard English?

Reviewer #1: Yes

Reviewer #2: Yes

6. Review Comments to the Author

Reviewer #1: The authors have shown a commitment to improving the manuscript after the initial review by providing thorough responses to the reviewers' comments and successfully incorporating their suggestions. From my perspective as a reviewer, I believe the revised manuscript is now suitable for acceptance and publication.

Reviewer #2: Reviewer #2: Ref paper: PONE-D-23-36127 R1

The authors prepared a modified version (PONE-D-23-36127R1) of the manuscript by Da Peng et al.

Taking into account the improvements noted in the modified version of the paper. I recommend this paper for publication in PLOS ONE.

7. PLOS authors have the option to publish the peer review history of their article (what does this mean?). If published, this will include your full peer review and any attached files.

Reviewer #1: No

Reviewer #2: **Yes: **Hamza DIF

---

## [Editor Report · Acceptance letter]

26 Jan 2024

PONE-D-23-36127R1 

PLOS ONE

Dear Dr. Peng, 

I'm pleased to inform you that your manuscript has been deemed suitable for publication in PLOS ONE. Congratulations! Your manuscript is now being handed over to our production team.

Kind regards, 

on behalf of

Dr. S. M. Anas 

Academic Editor

PLOS ONE